# Potential Protective Mechanisms of Ketone Bodies in Migraine Prevention

**DOI:** 10.3390/nu11040811

**Published:** 2019-04-10

**Authors:** Elena C. Gross, Rainer J. Klement, Jean Schoenen, Dominic P. D’Agostino, Dirk Fischer

**Affiliations:** 1Division of Paediatric Neurology, University Children’s Hospital Basel (UKBB), University of Basel, 4056 Basel, Switzerland; dirk.fischer@ukbb.ch; 2Department of Radiation Oncology, Leopoldina Hospital Schweinfurt, 97422 Schweinfurt, Germany; rainer_klement@gmx.de; 3Headache Research Unit, University of Liège, Dept of Neurology-Citadelle Hospital, 4000 Liège, Belgium; jschoenen@uliege.be; 4Department of Molecular Pharmacology and Physiology, Metabolic Medicine Research Laboratory, Morsani College of Medicine, University of South Florida, Tampa, FL 33612, USA; ddagosti@health.usf.edu; 5Institute for Human and Machine Cognition, Ocala, FL 34471, USA

**Keywords:** migraine, beta-hydroxybutyrate, ketone bodies, ketosis, migraine prevention, ketogenic diet, exogenous ketone bodies

## Abstract

An increasing amount of evidence suggests that migraines are a response to a cerebral energy deficiency or oxidative stress levels that exceed antioxidant capacity. The ketogenic diet (KD), a diet mimicking fasting that leads to the elevation of ketone bodies (KBs), is a therapeutic intervention targeting cerebral metabolism that has recently shown great promise in the prevention of migraines. KBs are an alternative fuel source for the brain, and are thus likely able to circumvent some of the abnormalities in glucose metabolism and transport found in migraines. Recent research has shown that KBs—D-β-hydroxybutyrate in particular—are more than metabolites. As signalling molecules, they have the potential to positively influence other pathways commonly believed to be part of migraine pathophysiology, namely: mitochondrial functioning, oxidative stress, cerebral excitability, inflammation and the gut microbiome. This review will describe the mechanisms by which the presence of KBs, D-BHB in particular, could influence those migraine pathophysiological mechanisms. To this end, common abnormalities in migraines are summarised with a particular focus on clinical data, including phenotypic, biochemical, genetic and therapeutic studies. Experimental animal data will be discussed to elaborate on the potential therapeutic mechanisms of elevated KBs in migraine pathophysiology, with a particular focus on the actions of D-BHB. In complex diseases such as migraines, a therapy that can target multiple possible pathogenic pathways seems advantageous. Further research is needed to establish whether the absence/restriction of dietary carbohydrates, the presence of KBs, or both, are of primary importance for the migraine protective effects of the KD.

## 1. Introduction

Migraine is a complex, common and debilitating neurological disorder [1]. Its episodic form is characterized by recurrent moderate to severe, typically throbbing and unilateral headache attacks that last between 4–72 h, which are aggravated by any kind of physical activity and accompanied by either photo-, phono-, or osmophobia, nausea, or a combination of these. Migraine affects approximately 17% of women and 8% of men in Europe [2], and with a peak incidence during the most productive years of life, migraine not only causes a huge amount of suffering, but also inflicts a substantial number of costs on society: approximately €18.5 billion per year in Europe alone [3,4]. Current migraine treatment options have limited efficacy and many—often intolerable—side-effects [5,6], with the potential exception of the very recent addition of monoclonal Calcitonin gene-related peptide (CGRP) antibodies [7]. Despite migraine’s primary pathogenic mechanisms being still largely unknown [8], accumulating evidence suggests that migraines could be—at least partially—an energy deficit syndrome of the brain, and the migraine attack a response to increased oxidative stress and/or (cerebral) hypometabolism [9]. Therapeutic approaches targeting cerebral metabolism may be warranted.

Ketone bodies (KBs: D-β-hydroxybutyrate (D-BHB), acetoacetate (AcAc), and to a lesser extent acetone) are mainly produced by the liver, but also other tissues, such as astrocytes [10], when glycogen storage is deprived, to serve as energetic substrates in the absence or severe reduction of dietary glucose, in particular for the heart and the brain. Mimicking this state of fasting, the ketogenic diet (KD) promotes the hepatic production of KBs with a high fat, low carbohydrate and moderate protein content. It was developed about 100 years ago after the observation that prolonged fasting has anticonvulsive properties [11]. Within recent years, the KD has received renewed interest, in particular since KBs might be beneficial for a variety of other neurological disorders [12,13,14]. All brain cells have the capacity to use KBs as respiratory substrates [10].

Out of the three physiological KBs, D-BHB constitutes up to 70% of KBs produced during ketosis [15] and is of particular interest, since it is not only a glucose transporter protein, i.e., a (GLUT)-independent alternative metabolite, but also a vital signalling molecule [16]. Many of these collateral effects make it a molecule of interest for therapeutic purposes. During a standard Western diet, the blood concentration of D-BHB is very low (<0.2 mmol/L) compared to glucose (≅ 5 mmol/L) [17]. During fasting or the KD D-BHB concentrations typically rise to levels between 0.5–5 mmol/L and up to 8 mmol/L during starvation [18]. Elevated KB levels have been shown to be well tolerated for extended periods of time (up to several years [19,20,21,22,23,24,25,26,27,28,29,30,31,32]).

Several case studies have demonstrated the potentially migraine protective effects of ketosis [22,33,34,35,36,37]. A one-month observational study of KD in 96 migraine patients as part of a weight loss program found a reduction of up to 80% in migraine frequency, severity and acute medication use [37]. The same intervention in 18 episodic migraineurs induced a 62.5% reduction in migraine days, which was accompanied by a normalization of the interictal habituation deficit of visual evoked responses [36]. The reduction in migraine attack frequency, severity and the use of acute anti-migraine medication during ketosis had effect sizes ranging from a total absence of attacks [33] to a reduction to 1/5th of the run-in period [37]. In addition, preliminary evidence suggests that the protective effect may outlast the duration of ketosis [33], as is often the case in pediatric epilepsy patients, and could be the result of longer-lasting gene-expression changes [12,38]. 

This review will describe the mechanisms by which the presence of ketone bodies, D-BHB in particular, could influence migraine pathophysiology (see Figure 1). To this end, common abnormalities in migraine (such as abnormalities in glucose metabolism and transport, mitochondrial functioning, oxidative stress, cerebral excitability, inflammation and the gut microbiome) are summarised with a particular focus on clinical data, including phenotypic, biochemical, genetic and therapeutic studies. Experimental animal data will be discussed to elaborate on the potential therapeutic mechanisms of elevated KBs in migraine pathophysiology with a particular focus on the actions of D-BHB. Please note that there is not enough research at present to disentangle the potentially differential effects of D-BHB within the scope of a KD (i.e., endogenous KBs via restriction of dietary carbohydrates) versus D-BHB added to a standard Western diet (i.e., exogenous BHB in addition to dietary carbohydrates). Research studies using either method will be cited, but not contrasted. 

## 2. Potentially Migraine Relevant Mechanisms of Ketosis

### 2.1. Hypoglycemia/Hypometabolism

Hypoglycaemia has been associated with migraine for almost a century [39,40,41] and fasting/skipping a meal is not only amongst the most commonly cited migraine triggers [42,43,44], but it can also be used experimentally to elicit migraine attacks in susceptible patients [39]. Due to very limited glycogen stores and high energy demands, the human brain is highly dependent on fuel sources from the circulation that can pass the blood-brain barrier and especially vulnerable to their potential short-comings. A simple comparison between migraine associated symptoms, premonitory symptoms in particular, and symptoms of hypoglycaemia [45] show several similarities: for example, dizziness, pale skin, cold hands and feet, binge eating/sugar cravings, yawning, nausea, low blood pressure, shaking, cognitive difficulties, tiredness, fatigue, visual dysfunction and slurred speech. Increased migraine frequency has also been observed during Ramadan [46] and migraine prevalence in type 2 diabetics was shown to proportionally increase with the number of hypoglycaemia attacks [47]. 

Further support for the role of energy deficiency and/or glucose metabolism in migraine comes from neuroimaging studies. Using 31P-MRS, an impairment of brain oxidative phosphorylation (OXPHOS) was detected first during migraine attacks [48] and thereafter between attacks [49,50,51,52,53,54,55,56]. OXPHOS abnormalities in patients with migraine were found both in the resting brain and in the muscle following exercise [49,56,57] (review by [54]), where a reduced glycolytic flux could be demonstrated [49,51]. More recently, a 16% decrease of absolute ATP levels in migraine without aura patients was demonstrated interictally using 31P-MRS [58]. This hypometabolism is generally found to moderately correlate with attack frequency [52,56,58]. A recent study comparing resting cerebral glucose uptake using 18-fluorodeoxyglucose-PET and visual cortex activation using visual evoked potentials showed that visual neuronal activation exceeded glucose uptake in visual areas in 90% of interictal migraine without aura patients, but in only 15% of healthy controls [59]. This supports the concept that a mismatch between brain activity and glucose metabolism may be a cornerstone of migraine pathophysiology. 

KBs are known to be able to counteract some of the negative effects of hypoglycemia and/or hypometabolism or prevent it all together. D-BHB has been shown to efficiently prevent neuronal death in the cortex of hypoglycemic animals and in vitro it was found to stimulate ATP production in glucose deprived cortical cultures [60]. Glycolysis is reduced in the presence of D-BHB and ketosis proportionally spares glucose utilization in the brain [61,62]. When present in sufficient concentration to saturate metabolism, D-BHB provides full support of all basal (housekeeping) energy needs and up to approximately half of the activity-dependent oxidative needs of neurons in 36h fasted rats [63]. Not only is D-BHB an alternative fuel source for the brain, it also seems to be more efficient. When catabolized for the synthesis of ATP in mitochondria, D-BHB produces more ATP per oxygen molecule consumed than many other respiratory substrates [64] and general “positive” shifts in energy balance have been observed [65,66,67].

There is circumstantial evidence from early experimental studies that oral glucose tolerance tests after an overnight fast can elicit migraine attacks in susceptible patients [68,69]. Interestingly, the metabolic responses in patients who developed an attack differed substantially from those that did not: free fatty acid (FFA) and KB levels increased significantly in the former, already before headache onset, and kept increasing despite similar food intake [68,69]. This can be interpreted as a counter-regulatory response to a cerebral energy deficit. Since KBs are an efficient alternative fuel for the brain, when glucose availability is low, their elevation would be expected to restore brain energy homeostasis, if present in sufficient quantity. 

### 2.2. Glucose Transport 

GLUTs are a wide group of membrane proteins that facilitate the transport of glucose across a plasma membrane. GLUT1 is an insulin-independent glucose transporter responsible for transporting glucose under basal conditions in all cells. This is especially the case in endothelial cells of the blood brain barrier as well as astrocytes and oligodendrocytes. In addition to glucose utilization, glucose transport might also play a role in migraine. GLUT1 deficiency syndrome has been linked to hemiplegic migraine and migraine with aura [70]. 

GLUT4 (adipose tissue and striated muscle) and GLUT3 (neuronal and glial cells) are the major insulin-mediated glucose transporters [71]. Insulin is the main anabolic hormone of the body and the key regulator of glucose homeostasis. It promotes the absorption of glucose from the blood and simultaneously blocks carnitine transporters and thus the penetration of FFA into the cells. Interictal impaired glucose tolerance and insulin resistance in migraine has been reported in various studies [72,73,74,75,76], but the evidence is not always conclusive [77]; see also the review by [78]. Some genetic support for a potential role of insulin in migraine comes from associations between polymorphisms in insulin-related genes and migraine [79,80,81,82].

KBs are taken up in to the brain via the monocarboxylate transporters (MCTs) and are hence completely GLUT and insulin independent. This KB transport mechanism allows the body and brain access to fuel even when glucose transport (GLUT1 or GLUT3/4) is compromised. Furthermore, KBs can also be produced endogenously in brain by astrocytes that have the capacity to metabolize free fatty acids and ketogenic amino acids L-Lysine and L-Leucine [10,83,84]. A strict KD can lead to complete remission in GLUT1 deficiency syndrome [85]. Additionally, a KD has been shown to lead to a marked upregulation of both GLUT1 and MCTs [86], thereby further enhancing available energy to the brain in a not completely GLUT1 compromised individual.

### 2.3. Mitochondrial Functoning 

Several lines of evidence point towards a role of mitochondrial functioning in migraine. The prevalence of migraine in mitochondrial disorders is more than doubled (29–35.5% of patients) [87,88] and migraine-like attacks in Mitochondrial encephalomyopathy, lactic acidosis, and stroke-like episodes (MELAS) are especially severe and prolonged [89]. Maternal transmission in migraine is more common [90], which suggests that either an X-linked form of inheritance could be involved or that mitochondrial DNA (mtDNA) transmission plays a role, since mtDNA derives exclusively from maternal origin. Furthermore, enrichment of a migraine genome-wide association study (GWAS) signal was found for mitochondria in both subcortical areas and the cortex (amongst others), a finding that identifies a genetic link between mitochondrial function and common migraine [91].

Further support for a generalized metabolic dysfunction in migraine comes from the reduced activity of mitochondrial enzymes, such as monoamine-oxidase, succinate-dehydrogenase, NADH-dehydrogenase, cyclooxygenase (COX) and citrate-synthetase in the platelets of migraine patients with and without aura [92,93]. Interestingly, these biochemical changes are restricted to enzymes of the respiratory chain that are encoded by mtDNA. In contrast to nuclear DNA, mtDNA is particularly sensitive to ROS because it lacks protection from histones [94,95].

Therapeutic studies also support mitochondrial dysfunction in migraine. Most nutraceuticals that have been demonstrated to be migraine preventative can directly be linked to energy metabolism and/or mitochondrial functioning, such as: high dose riboflavin (200–400 mg) [96,97,98,99,100], coenzyme Q10 (400 mg capsules or 300 mg liquid suspension) [101,102,103,104,105,106], alpha-lipoic acid (or thioctic acid) [107,108,109], B vitamins [110,111,112] and magnesium [113]. Even pharmaceutical prophylactic agents used against migraine are able to influence mitochondrial functioning and metabolism. For example, Topiramate prolongs mitochondrial survival, increases the activity of the mitochondrial respiratory chain complex [114], protects against oxidative stress, inflammation [115] and mitochondrial membrane depolarization, and has an insulin-sensitizing effect on adipocytes in female rats [116]; Amitriptyline also increases antioxidant capacity and reduces markers of oxidative stress [117] and Valproate preserves mitochondrial function in a rat model of migraine [118] and increases mitochondrial biogenesis [119].

KBs have been shown to enhance mitochondrial function [120,121,122] p. 20, [123] and stimulate mitochondrial biogenesis in the rat [65,124,125]. Furthermore, D-BHB may bypass complex I deficits due to its effects on complex II (succinate dehydrogenase) [123], thereby maintaining mitochondrial respiration and ATP production even in the presence of a complex I inhibitor (rotenone), but not a complex II inhibitor (malonate) [126].

### 2.4. Oxidative Stress 

While reactive oxygen species (ROS) and reactive nitrogen species (RNS) are necessary for certain signalling pathways, their unregulated production is deleterious, if it exceeds the anti-oxidant capacity of the organism. All common migraine triggers (stress/relaxation thereafter, fasting/skipping a meal, sleep changes (too much or too little), hormonal changes (including menses or oral contraceptives), weather changes (including hypoxia and high altitude), physical exercise (including sexual activity), alcohol, strong odours (especially perfume or cigarette smoke), intense light (especially bright or blue light) and loud noises [42,43,127]) are likely to negatively impact the balancing of oxidative stress levels, either directly or indirectly via negatively impacting mitochondrial functioning or energy metabolism [9,128].

Free iron is highly pro-oxidant and accumulates in the brain stem nuclei of migraine patients proportionally to disease duration [129]. Other heavy metals with pro-oxidant properties may also be increased in migraine [130]. Increased oxidative/nitrosative stress and/or decreased anti-oxidant capacity have also directly been found in migraine patients [117,131,132,133,134,135,136,137,138,139,140,141,142,143]. Of all biomarkers examined, superoxide dismutase (SOD) activity seems to be consistently reduced in migraine patients, also interictally [144]. Reduced anti-oxidant capacity or increased oxidative stress in migraine could be related to a genetic predisposition. A polymorphism in the SOD2 gene was associated with unilateral cranial autonomic symptoms in migraine with aura patients [145], and in paediatric migraine patients’ polymorphisms in the SOD2 and catalase gene were significantly higher in both migraine with and without aura patients compared to controls [146].

Further support for the role of oxidative stress in migraine is that as aforementioned antioxidants, such as Coenzyme Q10 (400 mg capsules or 300 mg liquid suspension) [101,102,103,104,105,106] and alpha-lipoic acid (or thioctic acid) [107,108,109] have been shown to have migraine protective effects.

The elevation of KBs, D-BHB in particular, seems to be an effective method for combating the negative consequences of elevated ROS/RNS. Systemic administration of D-BHB has been shown to reduce ROS production in distinct cortical areas and subregions of the hippocampus and efficiently prevented neuronal death in the cortex of hypoglycemic animals [60]. KBs themselves (AcAc, D-BHB and the non-physiological isomer L-BHB) have scavenging capacity [147].

Hydroxyl radicals (*OH) were effectively scavenged by D- and L-BHB, but only the administration of D- or L-BHB, but not of AcAc, was able to prevent the hypoglycemia-induced increase in lipid peroxidation in the rat hippocampus [147]. Furthermore, the metabolism of KBs results in a more negative redox potential of the NADP antioxidant system, which is a terminal destructor of ROS [148]. By increasing NADH oxidation, KBs have also been shown to be able to inhibit mitochondrial production of ROS following glutamate excitotoxicity [149].

D-BHB is a natural inhibitor of class 1, 2a, 3 and 4 histone deacetylases that repress transcription of the FOXO3a gene; this epigenetic action results in transcription of the enzymes of the antioxidant pathways, such as mitochondrial superoxide dismutase (MnSOD), catalase and metallothionein [150,151,152,153,154]. This has been shown to lead to significantly reduced markers of oxidative stress, such as lipid peroxidation and protein carbonylation, in the kidneys of D-BHB-treated compared to control mice [153]. BHB was also shown to increase FOXO3a activity through direct AMPK phosphorylation [155]. Furthermore, up-regulated glutathione and lipoic acid biosynthesis, enhanced mitochondrial antioxidant status, and protection of mtDNA from oxidant-induced damage in rats fed a KD for 3 weeks compared to controls have also been demonstrated [156]. One possible mechanism by which glutathione biosynthesis may be increased is through the activation of the Nrf2 transcription factor pathway [121]. Hence, KBs may increase antioxidant capacity via several mechanism [121,150].

To summarise, KBs, D-BHB in particular, generate lower levels of oxidative stress and increase antioxidant protein levels in combination with a higher cellular energy output [16,121,150]. D-BHB has previously been shown to be elevated as a compensatory response against oxidative stress in failing mice hearts [152]. It can be hypothesized that the aforementioned increased KB levels during a migraine attack [68,69] could represent an analogous reaction in response to increased oxidative stress. 

### 2.5. Cerebral Excitability 

The KD is known to be effective as treatment for refractory epilepsy [157]. There is comorbidity and shared genetic susceptibility between migraine and epilepsy [158] and hence partially similar pathophysiologic mechanisms are not unlikely. This assumption is supported by the overlap in certain pharmacological agents used to treat both disorders. One of the most reproducible and ubiquitous interictal abnormalities of the migraineurs’ brain is a lack of habituation in neuronal information processing [159]. While the exact underlying mechanisms of this phenomenon are uncertain, an imbalance between neuronal activation and inhibition is likely to be a pathogenic cornerstone in migraine, as it is in epilepsy [160,161,162,163,164]. The genetic association between migraine and epilepsy is illustrated by their co-occurrence in the hitherto known three subtypes of familial hemiplegic migraine (FHM), a rare autosomal dominant monogenic form of migraine. The three mutated genes are responsible for increased glutamate release (FHM1—CACNA1A) [165,166], reduced glutamate re-uptake (FHM2—ATP1A2) [167] or a decreased excitation of inhibitory interneurons (FHM3—SCN1A) [168]. In the common forms of migraine with and without aura, GWAS have identified 38 susceptibility loci [169,170,171]. These loci point towards genes involved in a variety of functions including glutamate release and re-uptake, and generation of action potentials [172], which may lead to a “generalized” neuronal hyperexcitability of the migraine brain. 

The observation that increased KB metabolism produces seizure protection suggests that fuel utilization and neuronal excitability are linked, however, the mechanisms underlying this link are poorly understood [173]. In addition to providing an alternative and more effective energy substrate, its positive effect on oxidative stress levels and mitochondrial functioning, some additional mechanisms have been proposed to underlie the effects of ketosis on cerebral excitability:

A higher synthesis of the inhibitory neurotransmitter GABA from glutamate [174] and a reduction of neuronal firing in GABAergic neurons of the substantia nigra pars reticulata have been demonstrated in response to KBs; this reduction being greater in faster-firing neurons [175]. Another effect of KBs on neural excitability seems to be mediated by an inhibition of glutamate transport and reduction in glutamate release [176], which in turn affects excitatory synaptic transmission [176]. Further reduction in excitability might be achieved via adenosine signalling through A1 purinergic receptors [177] and regulation of excitability by the activity of lactate dehydrogenase [178]. D-BHB has also been shown to be an agonist at FFA receptor GPR41, directly modulating the activity of N-type Ca^2+^ channels [179]. Furthermore, a KD was shown to activate inward rectifying potassium channels (metabolically sensitive K(ATP) channels) that in turn stabilize central neural excitability [38,175,180,181]. D-BHB specifically can bind to BAD (BCL-2 agonist of cell death) that, in addition to being pro-apoptotic, disrupts glucose metabolism KBs and hence opens K(ATP) channels [181]. An increase in inhibition as well as a reduction in excitability could be migraine protective.

### 2.6. Cortical Spreading Depression

Cortical spreading depression (CSD) denotes a wave of cellular depolarization of the neurons and neuroglia within the cerebral cortex and has been implicated as the underlying pathophysiological mechanism in migraine aura. CSD susceptibility is strongly modulated by metabolic factors. Hypoxia can trigger a CSD [182,183] and cerebral glucose availability modulates extrinsically induced CSD in both directions [184,185]. Hypoglycaemia significantly prolongs CSD duration and hyperglycaemia protects the tissue from CSD induction [184]. Supplying the rat brain with an alternative energy substrate to glucose via both short- and long-term treatment with a middle chain triglyceride enriched ketogenic diet has a similar protective effect against CSD [21]. 

### 2.7. Inflammation

Inflammation is a localised response designed to protect tissues against disease, infection or injury. Even though the involvement of neurogenic inflammation in migraine remains controversial [186], and migraine has not classically been considered an inflammatory disease, possibly because it is not obviously associated with redness, heat and swelling, several lines of evidence point towards the involvement of pro-inflammatory peptides or a “sterile neurogenic inflammation” in migraine pain (review [187,188]). Most importantly, calcitonin gene related peptide (CGRP) [189,190,191,192,193] see review [194], but also substance P, vasoactive intestinal peptide (VIP), pituitary adenylate cyclase-activating polypeptide (PACAP), nitric oxide (NO) [135,195,196,197,198] and to some extent cytokines [199], are all molecules associated with migraine both in animal and human studies review [187,200]. Cytokine Polymorphism (TNF-α and IL-1β gene polymorphisms) in patients with migraine without aura provide some more suggestive evidence for a possible contribution of inflammation in migraine [201]. Meningeal mast cell activation has been discussed to play a role in meningeal nociceptor activation in migraine [202]. 

D-BHB was reported to reduce inflammation generated by macrophages via a mechanism independent from any of those reported above [203]. The NLR family, pyrin domain-containing 3 (NLRP3) inflammasome, which is expressed mainly in immune cells, is activated by a shift to low cytoplasmic K+ levels [204]. Youm et al. noted that D-BHB prevented the decline in K+ levels and prevented activation of NLRP3 [203]. A reduction in pain and inflammation has been observed in rats fed the KD [205,206,207], which could further be migraine protective.

### 2.8. Gut Microbiome

The potential role of the microbiome in migraine pathophysiology is not yet fully established. Two recent reviews report an increased frequency of gastrointestinal (GI) symptoms or disorders in migraine patients compared to the general population, such as increased rates of nausea, cyclic vomiting syndrome, inflammatory bowel syndrome, irritable bowel syndrome, celiac disease, gastroparesis, hepatobiliary disorders, helicobacter pylori infection, gastric stasis, and alterations in the microbiota [208,209,210]. While there was no evidence for an added benefit of probiotics in an RCT with 63 episodic migraineurs [211] an uncontrolled observational study with 1020 patients using a multispecies probiotic showed a significant reduction in migraine days, headache intensity and the use of painkillers [212]. Additionally, an immunoglobulin-G based elimination diet among migraine patients with irritable bowel syndrome was associated with significant reductions in attack frequency, duration, severity and medication use [213]. Possible underlying mechanisms of migraine and GI diseases could be inflammation, alterations in the gut microbiota and its metabolites (such as neurotransmitters and neuropeptides) as well as increased gut permeability. 

The role of inflammation in migraine as well as the anti-inflammatory effects of ketosis have already been reviewed above. In addition, it has very recently been shown that a KD in children with severe epilepsy alters the gut microbiome [214,215]. Recent experiments in mice suggest that the KD seems to mediate some of its anti-seizure effects [216]. Mice treated with antibiotics or reared germ free were resistant to KD-mediated seizure protection otherwise found. Enrichment of and co-colonization with the KD-associated *Akkermansia* and *Parabacteroides* was shown to restore seizure protection. Even in mice fed a control diet, transplantation of the KD gut microbiota and treatment with *Akkermansia* and *Parabacteroides* each conferred seizure protection [216].

In support of this, in infants with refractory epilepsy a KD decreased seizure frequency in the majority of cases and significantly changed the gut microbial composition towards that of healthy controls: *Bacteroides* and *Prevotella* increased, while *Cronobacter* levels decreased by approximately 50% [217]. Other studies show a normalization of the microbiota in two other neurological disorders, namely autism [218] and multiple sclerosis [219]. In the BTBRT+tf/j mouse model of autism spectrum disorder, a KD significantly increased the Firmicutes/Bacteroidetes ratio which is typically low in autism spectrum disorder and also normalized the overabundance of the mucin-degrading bacterium *Akkermansia muciniphila* [218]. In patients with multiple sclerosis who were characterized by a reduced mass and diversity of microbial species, a KD initially decreased total gut bacterial concentrations during the first weeks, but, when maintained over six months, was able to restore the microbial mass to levels similar as in healthy controls [219].

However, these findings could not be replicated in all studies. For example, a small study on children with severe epilepsy found that the relative abundance of *bifidobacteria* as well as *E. rectale and Dialister* was significantly diminished during the KD intervention while an increase in relative abundance of E. coli was observed instead [214]. In a small study on GLUT1 deficiency syndrome a significant increase in *Desulfovibrio spp*., a bacterial group supposed to be involved in the exacerbation of the inflammatory condition of the gut mucosa, was found [220]. The differences observed might be due to the kind of KD employed, as different forms of the KD can vary tremendously in micro- and even macronutrient content, depending on the ratio of fat to protein/carbs, the use of processed meal replacements or processed vegetable oils, source of protein and carbohydrates and other factors. The children with severe epilepsy were mostly on a very strict classical KD with 4:1 ratio that allows for little fibre content. In addition, the use of pro-inflammatory processed vegetable oils and instant ketogenic formulas or meal replacements may be suboptimal for the gut microbiome. Lastly, given the structural and functional similarity between butyrate and BHB, it could be hypothesized that higher systemic concentrations of the latter in case of the higher KD rations (4:1) could decrease the importance of microbial butyrate production [221].

In sum, an alteration of the microbiome and its downstream beneficial effects on gut permeability, synthesis of metabolites and neuropeptides, as well as inflammation could thus be another potential disease modifying mechanism of ketosis in migraine. 

## 3. Discussion and Conclusions

Migraine is a very heterogeneous disease, with most probably a multitude of fairly common genetic polymorphisms and in turn pathophysiological mechanisms contributing to the migraine phenotype. We have reviewed the potential contribution of eight such pathophysiological mechanisms and their possible exploitation through dietary ketosis (KDs and/or D-BHB supplementation): (1) hypoglycaemia/hypometabolism, (2) glucose transport, (3) mitochondrial functioning, (4) oxidative stress, (5) cerebral Excitability, (6) CSD, (7) inflammation and (8) the microbiome. Which mechanisms contribute to the migraine phenotype in a given individual are likely to vary. Further research is needed to confirm these mechanistic hypotheses and their translational relevance for patients. Furthermore, the scope and importance of these metabolic mechanisms within the broad range of migraine pathophysiology remains to be determined.

With migraine being such a diverse and multigenic disease, finding the one treatment target seems a nearly impossible endeavour. Exciting recent technological advances in the field of genetics and induced pluripotent stem cells are paving the way for future more personalised treatment approaches. Until those reach the clinic, an elevation of KBs, D-BHB in particular, which have been shown to potentially influence all of the aforementioned migraine pathophysiological mechanisms, might offer a long-needed relatively side-effect free remedy for at least a proportion of migraine sufferers. 

It remains to be determined whether the absence/restriction of dietary carbohydrates, the presence of KBs, or both, are of primary importance for the potentially migraine protective effects of the KD that has previously been demonstrated. Additionally, third factors, such as increased fatty acids, amino acids, supplementation with medium-chain triglycerides [222] or other dietary changes as well as alteration of the microbiome could also be disease modifying. The potential preventative anti-migraine effect of supplementation with BHB without a strict dietary change is currently being examined in an RCT [223] and could help answer some of these questions.

Moreover, a lot of the mechanistic effects of ketosis and/or presence of BHB have been examined in animals and more clinical research is needed to validate those effects. Such future clinical research could additionally help determine whether, and to what extent all the aforementioned potentially disease-modifying effects of ketosis are actually also occurring in migraine patients.

## Figures and Tables

**Figure 1 nutrients-11-00811-f001:**
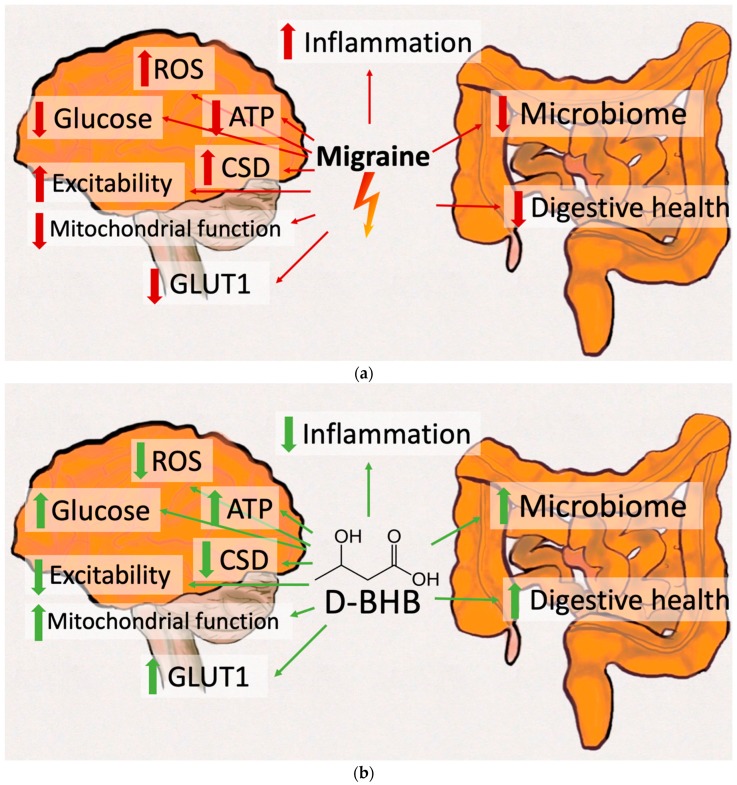
Potentially migraine relevant mechanisms of ketosis. (**a**) Amongst key migraine pathophysiological mechanisms are hypometabolism, decreased glucose transport (including glucose transporter 1 (GLUT1) deficiency), reduced mitochondrial functioning, increased cerebral excitability, increased cortical spreading depressions (CSD) incidence, increased oxidative stress (reactive oxygen species (ROS)), increased inflammation, microbiome abnormalities and reduced digestive health. (**b**) D-β-hydroxybutyrate (D-BHB; with or without the context of a ketogenic diet) has been shown to positively influence each of these mechanisms: increasing cerebral metabolism, increasing glucose transport (including glucose transporter 1 (GLUT1) deficiency), increasing mitochondrial functioning, reducing cerebral excitability, decreasing cortical spreading depressions (CSD) incidence, reducing oxidative stress (reactive oxygen species (ROS)), decreasing inflammation, improving the microbiome and increasing digestive health. ATP = adenosine triphosphate; CSD = cortical spreading depressions; D-BHB = D-β-hydroxybutyrate; GLUT1 = glucose transporter 1; ROS = reactive oxygen species.

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
