# Peer review of "Potential Protective Mechanisms of Ketone Bodies in Migraine Prevention"

_nutrients, 2019, doi:10.3390/nu11040811_

Reviewer 1 Report

The present manuscript is a narrative review of the potential beneficial role of ketones and ketogenic diets in migraine disorders. The authors provide a summary of mechanisms potentially linking ketosis and migraines, and they present two figures as an overview of the pathological processes involved in migraines and their potential mediation by ketone bodies (particularly D-β-hydroxybutyrate).

This reviewer’s knowledge of the relationship between ketone bodies and brain metabolism is extremely limited. Therefore, I do not feel qualified to comment on the comprehensiveness of the literature review, and whether there are any obvious gaps in coverage. Having said that, I found this to be a well-written, logically organized review paper that also seems very informative. As such, I have no further suggestions.

Author Response

Dear reviewer,

many thanks for your positive  feedback. We appreciate it!

Many thanks and kind regards,

Elena Gross

Reviewer 2 Report

The review focuses on migraine highlighting the putative pathways that are altered during the disease that includes mitochondrial functioning, oxidative stress, cerebral excitability, inflammation and the gut microbiome. The author's determined in their review of how KBs could be potential therapeutics against migraine.

This review is nice, but it would be more intuitive if the authors provide main news and noteworthy about this article to make it unique among other previous literature sought in the field.

The issues in the article are-
Line 18-30: This line should be rearrange in such a way that the abstract section should able to explain the what you really review here main focus/highlight/novelty (Personalized statement).
Based on the  materials in the  main body,  please explain in the abstract section how you going to recapitulate.
" Line 80-91", is also needed to summarize in abstract section as well. So please make necessary changes and cut the text as necessary in the abstract.

Line 43: If CGRP are used for the first time, please give the full form.
Line 71: "Di Lorenzo et al., 2017, submitted" . I think submitted is not needed !
Line 151:  Is it increasing or decreasing the role of mitochondria?
Line 160 : check the spelling of mitochondria
Line 203-204: How other heavy metal reach to the brain crossing BBB during migraine, please highlight here.
Line 37 and 214: Please provide appropriate spacing here and elsewhere: Example: "72h", "400mg" or so on.
Line 269: What is faster firing neuron? Are they basket cells or others?
Line 308: "2.6 Gut Microbiome". Check the subheading number.
Line 281-287: This paragraph can be expanded.
Line 357: Discussion and Conclusion section is relatively short and this part has to be extended with more
critical discussion based on your main findings that this review brought out for advancing the field as compared to other previous report.
 (Similarity, discrepancy, scope, functional human relevance, caveats in the study can be the topic of interest for other to discuss).

Line 422-425: Here and elsewhere (if any), please make sure that references follows the journal's guideline.

Author Response

Dear Reviewer,

please find our answers in the PDF.

Many thanks,

Elena Gross
